# Experimental Research on Wolfram Inert Gas AA1050 Aluminum Alloy Tailor Welded Blanks Processed by Single Point Incremental Forming Process

**DOI:** 10.3390/ma16196408

**Published:** 2023-09-26

**Authors:** Gabriela-Petruța Rusu, Radu-Eugen Breaz, Mihai-Octavian Popp, Valentin Oleksik, Sever-Gabriel Racz

**Affiliations:** Faculty of Engineering, “Lucian Blaga” University, 550024 Sibiu, Romania; gabriela.rusu@ulbsibiu.ro (G.-P.R.); mihai.popp@ulbsibiu.ro (M.-O.P.); valentin.oleksik@ulbsibiu.ro (V.O.); gabriel.racz@ulbsibiu.ro (S.-G.R.)

**Keywords:** wolfram inert gas, incremental forming, single point incremental forming, aluminum alloy, strain, thickness reduction, forces, welded blanks, industrial robot

## Abstract

The present paper aims to study the behavior of tailor welded blanks subjected to a single point incremental forming (SPIF) process from an experimental point of view. This process was chosen to deform truncated cone shapes of AA1050 aluminum alloy with different thicknesses. A uniaxial tensile test was performed to determine the mechanical characteristics of the alloy. Initial experimental tests implicated the use of variable wall angle parts which were processed on unwelded sheet blanks for determination of the behavior of the material and the forming forces. Afterwards, the wolfram inert gas (WIG) welding technique was used for joining two sheet blanks with different thicknesses either through one pass on one side, or by one pass on both sides. The conclusion of this paper indicates that one-sided welded blanks cannot be deformed successfully without fracture. In case of two-sided welded blanks, the results showed that the desired depth of 25 mm can be reached successfully. In case of the SPIF process, if welded blanks must be deformed, then the suitable method is to weld the blanks on both sides.

## 1. Introduction

The process of single point incremental forming (SPIF) is gaining more and more popularity among researchers worldwide due to its significant potential in manufacturing complex components [1,2], characterized by reduced tooling, cycle time, and costs, as well as the ability to create sculptured profiles, as compared to traditional methods [3]. The main advantages of the single point incremental forming process consist of using tools with simple shapes and the absence of an active die representing the conjugate shape of the part [4,5], as can be seen in Figure 1. However, it is essential to acknowledge certain limitations of this process. Notably, SPIF tends to be slower compared to other methods of plastic deformation such as deep drawing, stamping, and bending [4], and it sometimes results in parts with less geometrical accuracy. Currently, its applications are predominantly confined to small-series batches and the prototyping sector, where the benefits of reduced tooling costs and greater geometric flexibility ca be fully realized. Furthermore, creating parts with high wall angles presents a considerable challenge, and achieving 90° sections is even more demanding [6]. Despite these challenges, the mechanics of SPIF are relatively straightforward [7]. The sheet blank is secured at the intersection between the active die and the retaining ring. In operation, a punch characterized by a semispherical or parabolic profile traverses a defined toolpath, manipulating the sheet blank to yield a component with intricate geometry [8,9]. This straightforward approach, coupled with ongoing research, holds promise for addressing the current limitations and expanding the application scope of SPIF in the near future. This process is typically carried out on thin metal sheets, up to 1 mm in thickness. Of course, when it comes to plastic materials, these can be easily deformed even with thicknesses of 3–5 mm. Regardless of the studied material, the thickness of the sheets is generally limited by the capacity of the equipment used for their deformation, meaning the maximum forces that this equipment can generate. Thus, the higher the mechanical strength of the base material, the smaller the thickness of the sheets that can be deformed [10,11].

The single point incremental forming of welded sheets emerges as a frontier development in the field of sheet metal forming, notably for the added benefits conferred by the joining process. However, there are not many studies to date that address the single point incremental forming of welded sheets due to the challenges of achieving the welds [12]. Considering that the single point incremental forming process is carried out on sheets, the welded joints must be made end-to-end. Therefore, for end-to-end welding of two sheets, I-shaped joints are used, and for this type of joint, as presented in the previous subsection, they are suitable for welding sheets with thicknesses between 1 and 14 mm. This makes it difficult to weld thin metal sheets used in the single point incremental forming process [13].

One of the foundational studies in the domain of single point incremental forming (SPIF) of welded sheets was spearheaded by Ambrogio et al. in 2006 [14]. Utilizing the friction stir welding process, the researchers adeptly joined 1.2 mm thick aluminum alloy sheets. This method, characterized by a tool engaging in a rotational and feed movement along the welding zone, fostered high temperatures and plastic flow phenomenon, culminating in a welded joint. Through meticulous optimization, parameters such as a tool rotation speed of 1040 rpm, feed rate of 104 mm/min, and a 1.1 mm tool penetration depth were established as optimal. Consequently, they successfully crafted a truncated pyramid featuring a 40 mm depth and a 50° wall angle, identifying 58.75° as the critical deformation angle before weld joint rupture. The study of single point incremental forming of welded sheets made from two different aluminum alloys, AA6061 and AA5083, was conducted by Tayebi and collaborators [15]. They investigated the process both experimentally and through numerical simulations carried out in ABAQUS. They successfully deformed the welded sheets of both alloys at an angle of 60° to a total depth of 89.75 mm. At angles of 62.5° and 65°, the sheets experienced failure at an early stage of the single point incremental forming process, reaching a maximum depth of 22 mm. Additionally, they conducted a microstructural analysis of the deformed parts. Another approach was to investigate the effect of SPIF process parameters on the forming behavior of TWBs using a simulation method. ABAQUS/Explicit was used as a simulation tool, and SPIF process parameters like feed, spindle rotational speed, and coefficient of friction were selected for investigation [16]. In their paper, Tucci and colleagues presented an integrated numerical model designed to simulate the single point incremental forming of thin sheets that were previously friction stir welded [17]. This approach is based on the interconnection of two models that simulate the consecutive processes. The first model incorporates input data related to the friction stir welding process, such as tool geometry, rotation speed, feed rate, and other parameters characterizing the plastic behavior of the material. Subsequently, the second model performs the actual single point incremental forming process [18]. Another approach is presented by Maji, who studied the deformability of AA5083 aluminum alloy sheets with a thickness of 1 mm that were friction stir welded [15].

Maji conducted experimental trials to determine the optimal welding parameters and concluded that excellent performance is achieved at a punch rotation speed of 2000 rpm, a feed rate of 100 mm/min, and a tool diameter of 9 mm. Subsequently, a line test was conducted to determine the deformation limit diagram for both the base material and the welded sheets. Maji observed that the deformability of the welded sheets significantly decreases compared to the deformability of the base material. This reduction is attributed to the welding process, which leads to a hardening of the base material, resulting in higher hardness and reduced elongation at fracture in the heat-affected zone.

In another study, Alinaghian [19] experimentally investigated the deformability of AA6061 sheets with a thickness of 2 mm that underwent single point incremental forming. In the initial stage, the researcher determined which forming direction (at 0°, 45°, and 90° relative to the sheet rolling direction) exhibited the best deformability. Analyzing the deformation limit diagram, it was concluded that the variant with a 45° forming direction was optimal. Alinaghian employed the response surface methodology to determine optimal parameters for the single point incremental forming process. The findings highlighted parameters leading to increased deformability, such as a tool rotation speed of 1600 rpm, a feed rate of 40 mm/min, and a penetration depth of 0.15 mm. Similarly, optimal parameters for minimal material thinning included a processing speed of 600 mm/min, a vertical step of 0.6 mm, and no punch rotation. These findings are especially pivotal in the metal processing industry, where innovations emphasize advanced welding and forming technologies. A pertinent study revealed forming angle limits at 60° for base metal and 57.5° for FSW sheets [20], complementing Alinaghian’s insights and underscoring the importance of recognizing deformability constraints for optimized material performance.

It is noted that the formability of the dissimilar aluminum alloy sheets FSW in the SPIF process proves to be comparable to that of the base materials, highlighting the positive influences of dynamic recrystallization on the welded zones [21]. These findings collectively underscore the significance of understanding material behavior and process parameters to optimize both single point incremental forming and friction stir welding processes, ultimately advancing the capabilities of modern metal fabrication techniques.

The majority of researchers in the field have employed friction stir welding to create welded sheets for studying the behavior of single point incremental forming in welded sheets [22]. This is also evident in the work of Carlone and colleagues, who analyzed the single point incremental forming behavior of AA6082-T6 sheets with a thickness of 2 mm that were friction stir welded [23]. They investigated five different combinations of welding parameters, including a punch rotation speed ranging from 1000 to 1400 rpm and a feed rate ranging from 40 to 100 mm/min. They compared these parameters for truncated cone-shaped pieces welded through TIG welding. They identified optimal parameters, in terms of maximum deformation depth before material failure occurred, as 1200 rpm for the rotation speed and 700 mm/min for the feed rate. Additionally, they found that the feed rate during the welding process had the most significant influence on the deformability of the welded pieces, with a feed rate of 100 mm/min resulting in weaker outcomes. Currently, the need for using welded sheets is growing, especially in the automotive manufacturing industry. This is highlighted by Merklein and colleagues in their paper, where they outline applications of welded sheets in the automotive construction industry [10]. These welded sheets are subsequently employed as semi-finished products for conventional plastic deformation processes like deep drawing. Additionally, they provide the possibility for single point incremental forming, offering an alternative to the current production process of reinforcement, support, and structural components in vehicles. Applications of welded sheets in the construction of a modern vehicle can be used at: Cross member, A-pillar, B-pillar, roof reinforcement, motor compartment rail, front and rear door inner.

In his paper, Campos [24], along with his collaborators, defines three distinct zones of single point incremental formed sheets: the central forming zone, the thermomechanically affected zone, and the heat-affected zone. They determined that mechanical properties remain constant in the first zone, gradually decreasing until the edge of the heat-affected zone, where values are similar to those of the base material. In their study, they used an AA6082-T6 aluminum alloy. Consequently, the authors incorporated an isotropic Swift flow law into their finite element analysis model for single point incremental forming, accounting for the work-hardening coefficient (n) and the plastic resistance coefficient (K). Regardless of the chosen process for creating welded joints, in the case of a properly executed weld, the weld pool is formed, which possesses mechanical properties superior to the base material in terms of strength, while also exhibiting lower plasticity. However, due to the nature of the welding process, determining the mechanical properties of the weld pool is not straightforward.

A method for investigating the behavior of the weld bead is proposed by Marathe [25] through numerical simulations of welded sheets between two aluminum alloys: AA5083-O with a low yield strength of 150 MPa, and AA6061 T6 with a higher yield strength of 280 MPa. Marathe conducted numerous numerical simulations using various yield strength values for the weld bead, ranging between the limits of the two materials: 160, 180, 200, 220, 240, and 260 MPa. Furthermore, Marathe investigated the effect of rotating the weld bead, which is influenced by its yield strength, and obtained displacements of up to 1.2 mm. Another approach by the author, in collaboration with Raut [26], was to investigate the effect of punch path direction on the weld bead formed between two other aluminum alloys, AA5754 H22 and AA5052 H32. They demonstrated that the punch position at the beginning of the toolpath plays an important role in the displacement of the weld bead. A novel method to study the single point incremental forming process of friction stir welded sheets was proposed by Silva, who employed an additional protective sheet made from a different material placed over the welded sheets. The purpose of this approach was to shield both the workpiece and the weld bead from the rotational effects of the forming tool. Ghadmode suggested in his research about formability of tailor-welded blanks (TWBs) made from aluminum alloys. He observes that the single point incremental forming (SPIF) process significantly reduces the forming depth of these TWBs compared to the base materials, and highlights brittle behavior in the welded region. The author suggests that post-TIG welding heat treatment could enhance the formability of these TWBs [27]. Silva conducted numerous experiments on truncated cone and truncated pyramid-shaped pieces using AA1050-H111 sheets with thicknesses of 1.5 and 2 mm. For the protective sheet, DC04 material with a thickness of 0.63 mm was utilized. As a result of these experiments, the author concluded that it is possible to produce parts through single point incremental forming of friction stir welded sheets using an additional protective sheet made from a different material than the base material [28].

In another study, Baharudin analyzed the forces involved in the single point incremental forming process at a specific point for friction stir welded aluminum AA6061-T6 sheets with a thickness of 2 mm [29]. The authors also employed the Taguchi method to analyze the input parameters in the single point incremental forming process, such as punch rotation speed, processing speed, vertical step, and punch diameter. The fabricated parts took the shape of truncated cones with a variable angle, ranging from 30° at the larger base to 80° at the smaller base of the truncated cone. As conclusions, the authors observed that as the punch rotation speed increases, the forces in the process decrease. Additionally, they found that the processing speed and the vertical step have the most significant influence on the forces exerted in the process.

The motivation behind this research is to explore the potential of producing thin welded sheets with thicknesses less than 1 mm using conventional welding techniques. In industries like automotive and aerospace, there is often a demand for components with varied thicknesses to meet specific structural and weight requirements. For instance, car door panels or aircraft wing sections might require varying thickness for an optimal strength-to-weight ratio. Considering that the majority of the researchers in the field use the friction stir welding technique, our investigation into traditional methods provides a distinctive approach. In the case of welded blanks subjected to the SPIF process, there are controversial and diverging results, showing in some cases that welded blanks cannot be deformed without fracture, but also in other cases that the parts were obtained without fracture. 

The aim of this paper is to study the behavior of TWBs welded through WIG processed by single point incremental forming and the influence of the welding bead over material formability. For this purpose, AA1050 aluminum alloy blanks were deformed initially into a variable wall angle frustrum cone, and afterwards, TWBs were deformed until material fracture occurred and the depth was determined. The TWBs were welded either through one pass on one side, or with one pass on both sides. In the case of two-sided welded beads, the results showed that it is possible to deform the aluminum alloy until 25 mm depth without fracture. 

Considering the purpose of this research, which is to study the behavior of tailor welded blanks (TWB) through wolfram inert gas (WIG), we chose to use sheet blanks out of the same material but with varying thicknesses. This choice stems from the fact that the majority of researchers in the field uses sheet blanks from the same material and thickness. Those familiar with the welding area are aware that welding aluminum alloy components is much more challenging than welding other materials such as steel, titanium, magnesium alloys, etc. In the following sections, we will outline the research methodology employed in this paper. 

## 2. Materials and Methods

In this research, we analyzed the influence of the welding bead on the incrementally formed aluminum alloy sheets. The first part of the experimental research consists in carrying out the tensile test in order to determine the mechanical characteristics. The second stage of this research was focused on the study of the deformability on the incrementally formed sheets with frustrum cone shape parts with variable wall angle. The last stage involves the incremental forming of the aluminum sheets welded through the WIG process.

### 2.1. Welding Methodology

The sheet blanks used in this research were made of AA1050 aluminum alloy, which is a cold laminated sheet (EN573-3:2009) [30]. This type of alloy exhibits high corrosion resistance and can be easily processed through plastic deformation processes, showing good ductility, which make it suitable for use in the automotive and aeronautic industries for car body parts where low weight products are in high demand. The chemical composition according to the standard presented above is shown in Table 1.

For the welding of the TWBs, the following technological parameters were used: thin wolfram electrodes with a diameter of 3 mm, a direct current power source of 135 A, direct polarity and argon as the inert gas. The sheet blanks used for welding had a dimension of 125 × 250 mm, summing a total area of the blanks of 250 × 250 mm with a thickness of 0.8 and 1 mm. The TWBs were welded on one side through a single pass and on two sides also through a single pass as in Figure 2. 

The welding of the TWBs was performed using the welding apparatus Fronius MagicWave 4000 AC/DC. This apparatus stands as an advanced and dependable solution for WIG welding operations within the experimental research outlined in this article. Its primary features and functions make it an indispensable tool for achieving high-quality results and exploring weld behavior under various conditions.

### 2.2. Uniaxial Tesile Test for AA1050 Specimens

For the study of the behavior of the welded blanks processed by single point incremental forming (SPIF), it was necessary to carry out uniaxial tests of the specimens in order to determine the mechanical characteristics. For this test, the specimens used were composed of the base material without weld. For the experimental determination of the engineering stress-strain diagram, a tensile test was conducted using the Instron 5587 machine. The machine is controlled through Bluehill 2 software, which enables features like automatic sensor calibration, system monitoring, and the ability to determine the engineering characteristic curves. This machine has a maximum loading of 300 kN and an adjustable test speed in the range of 0.001–500 mm/min. Consequently, six rectangular cross-section specimens were tested for each thickness, 0.8 and 1 mm in the rolling direction. The specimens employed adhere to the SR EN 10002-1:2002 standard [31], possessing a calibrated length of 75 mm and a width of 12.5 mm.

The specimens were cut with the help of a water jet from sheets of AA1050 aluminum alloy. By using this method of cutting the specimens, the possibility of the measured characteristics being influenced by strong heating and even structural alterations in the material which can occur as a result of laser cutting are eliminated. The water jet cutting process was performed using the Digital Control cutting machine. During the tensile test, both ends of the specimen were clamped, and they were deformed at a constant rate until fracture occurred. Through this test, there were determined the following mechanical characteristics: elasticity modulus (Young’s modulus) E, yield strength (σ_c_), ultimate tensile stress (σ_max_), strain hardening coefficient (n), plastic resistance coefficient (K), and tensile strain at break (ε_max_). 

### 2.3. Single Point Incremental Forming Process

Considering the purpose and objectives of this paper, which involve studying the behavior of TWBs deformed through the SPIF process and analyzing the results obtained regarding deformability of the parts, the setup presented in Figure 3 was employed containing the following equipment: fixing system for sheet blanks, KUKA KR210-2 industrial robot, force transducer for measuring forces during process in three directions, fixing system for the punches, and an optical strain analysis system ARAMIS.

The fixing system for sheet blanks was composed of upper and lower retaining plates (1), a support for the retaining plates (2), a T-slot plate (3), a frame (4) secured to the floor, and fastening bolts (5), as can be seen in Figure 4.

The inner surface of the retaining plates measures 200 × 200 mm, ensuring a secure grip of the TWBs. The sheets are fastened by tightening the retaining plates using 12 M10 × 30 cylindrical-headed screws with hexagonal recesses. This setup was designed and constructed in a manner that allows the TWBs to be mounted vertically, facilitating strain measurements during the SPIF process using the ARAMIS 2M software.

The technological equipment used for sheet metal forming was the KUKA KR210-2 industrial robot. We chose this equipment due to its capability to allow the measurements of strains and its ability to follow complex toolpaths due to its high flexibility. The robot is a serial type with 6 kinematic joints, providing 6 degrees of freedom, which gives it a higher flexibility compared to general CNC machines. The robot’s maximum payload is 210 kg, allowing it to apply forces of up to 2 kN. This capacity is more than sufficient since the chosen aluminum alloy, which has a thickness of 0.8 and 1 mm, does not typically require such high forces, based on the bibliography study.

To measure the forces that occur during SPIF process, a piezoresistive force transducer from PCB Piezotronics, model 261A13 was utilized. This transducer has a capacity of 70 pF for all 3 channels and can measure maximum forces of up to 17.7 kN in the X and Y directions, and 44.48 kN in the axial Z direction (which aligns with the axis perpendicular to the sheet blanks). The sensitivity of the transducers is 7.19 pC/N for X and Y axes and 3.37 pC/N for the Z axis. The output signal from the piezoresistive transducer is analog, with values in a low-voltage range (mV). Consequently, in order to be measured and sampled, the signal needs to be amplified beforehand.

Signal amplification is achieved through a digital charge amplifier, HBM PACEline CMD 600 model. This amplifier generates an output signal in the range of 0–10 V, which is then transmitted to the data acquisition system, HBM QuantumX model MX840B. This system features an 8-channel data acquisition board, capable of measuring signals from strain gauges connected in half-bridge or full-bridge configurations, as well as piezoresistive, piezo capacitive, or piezo inductive transducers in full-bridge configurations. The system has a sampling frequency of up to 2400 Hz concurrently across all 8 channels. In this case, a sampling frequency of 50 Hz was used for the 3 force components measured during the SPIF process.

The clamping system used for the toolholder is depicted in Figure 5 and consists of a toolholder (1), the force transducer (3), and 2 connecting flanges (2 and 4) between the toolholder and force transducer, and between the force transducer and the robot’s end effector, respectively. The toolholder incorporates an ER32 collet chuck with a diameter of 10 mm and a nut for securing the collet (standardized for CNC machines, following ISO 30 specifications). The fixation of the flanges to the transducer is achieved using 8 screws with a 3/8” UNF thread.

For analyzing the behavior of the AA1050 aluminum alloy TWBs, the experiments involved frustrum cone shape parts with the following dimensions: diameter of the initial large base of 85 mm, total depth of 25 mm, and a wall angle of 55°. For the planning of the experiments, we decided to vary the technological parameters such as the vertical step and punch diameter in order to assess their impact on the deformability of AA1050 TWBs welded through the WIG process for different combinations of thicknesses. 

Considering the selection of three factors with three levels of variation each, conventionally, all possible combinations should be tested: 27 experiments. This would imply a large number of experiments and a lot of data to be analyzed, which makes it hard to grasp the behavior of TWBs during the SPIF process. A good approach in this case is to use a statistical method in order to reduce the number of experiments. In this paper, we applied the Taguchi method, and thus the number of experiments dropped to only 9 experiments (by using the L9 orthogonal array), which will yield relevant results for studying the deformability of AA1050 TWBs welded by WIG. Table 2 presents the 9 experiments conducted within this article.

The toolpath used for all experiments was a spatial spiral with the accordingly vertical step for each case, due to the fact that this toolpath showed previously in other research that it does not produce high local deformation and the strains are uniformly distributed around the wall of the parts. Due to the nature of the SPIF process being a slow process with high production time, the movement of the robot was 2400 mm/min, a value established in accordance with the ARAMIS 2M software, which acquires images at specific points in time. Thus, for a precise measurement, we used the acquisition of images every 4 s. One should note that while it is possible to acquire images faster, the file saved by the software becomes very large, so a sacrifice must be made. Before the experimental tests, the TWBs were sprayed with a white matte paint, and afterwards, a layer of dark points was sprayed onto the white sheet blanks for the optical strain analysis system to be able to measure the major and minor strains, as well as the thickness reduction. During the tests, the TWBs were lubricated with a thin layer of synthetic oil.

The toolpath was generated with the help of a CAM software, namely SprutCAM version 15, where the model of the robot was introduced as well as the model of the fixing system for the TWBs. Thus, a proper simulation of the process, which takes into the account the movement possibility of the robot and the space around it in order not to hit the fixing system, was possible. After the successful generation of the toolpath, it was postprocessed for the KUKA programing language and transferred to its controller.

## 3. Results and Discussions

Aluminum alloys can be welded through a process that provides enough heat to create a weld pool, and the WIG process is recommended for this operation. The use of pure argon as a shielding gas and the appropriate wire for the specific alloy are also essential in this process. For a better comprehension of the behavior of AA1050 welded sheet blanks via TIG welding, the experiments were carried out in three stages. During the initial stage, uniaxial tests were performed; in the subsequent stage, the behavior of non-welded sheet blanks under incremental formation was examined; and in the final stage, the conduct of sheet blanks welded through WIG was analyzed. The ensuing sections will showcase the outcomes derived from these experiments.

### 3.1. Uniaxial Test

After conducting the uniaxial test on the specimens obtained from the un-welded sheet blanks, utilizing the Bluehill 2 software, the following mechanical characteristics were obtained: Young’s modulus (E), yield strength (σ_c_), ultimate tensile strength (σ_max_), strain-hardening exponent (n), plastic resistance coefficient (K), and tensile strain at break (ε_max_), as shown in Table 3. The statistical analyses conducted included calculation of mean, median, standard deviation, coefficient of variation, and *p*-value. Through the Anderson-Darling test—a derivative of the Kolmogorov-Smirnov test—the normality of data distribution was assessed. The *p*-values, calculated using Minitab V19, confirmed the normality hypothesis for the AA1050 sheet blanks with thicknesses of 0.8 mm and 1 mm, as they were below the threshold of 0.05.

It was observed that in the case of AA1050 sheet blanks with a thickness of 0.8 mm, the average value of Young’s modulus for the specimens cut in the rolling direction is 66.21 MPa, while for a thickness of 1 mm, it is 69.49 MPa, showing a difference of 4.96%. The maximum yield strength is encountered in the specimens with a thickness of 1 mm, at 111.83 MPa, with a difference of 21% between the two thicknesses. In the case of ultimate tensile strength, the difference is 22%. The difference between the maximum strain-hardening exponent value for the specimens with a thickness of 0.8 mm and those of 1 mm is 27%. As well known, a lower strain-hardening coefficient value indicates a more heterogeneous internal structure. The plastic resistance coefficient follows the same trend, with the maximum values found for the 1 mm thickness, differing by 28%. The elongation of the aluminum alloy is relatively small. It can be observed that Young’s modulus decreases as the thickness of the sheet blanks increases. In Figure 6, the conventional stress-strain curves obtained from conducting the tensile test for the two thicknesses of the analyzed material are presented.

Further analysis revealed an improvement in the properties of the studied materials as the material thickness increases. This can be explained by the fact that greater thickness provides higher material strength and an increased capacity to withstand the forces and deformations imposed by the SPIF process. Thus, the thickness of the sheet blank must be chosen in a way that the material strength does not exceed the technological capabilities of the equipment used.

### 3.2. Single Point Incremental Forming of Non-Welded Sheets

In this subsection, a series of experiments are presented to evaluate the behavior of non-welded sheet blanks under incremental formation. The goal is to draw conclusions regarding the threshold values of process parameters that prevent the manufactured parts from fracture. For these experiments, three factors were varied to analyze their effects: combinations of thicknesses (0.8–0.8 mm, 0.8–1 mm, and 1–1 mm), vertical step (0.25 mm, 0.5 mm, and 0.75 mm), and punch diameter (6 mm, 8 mm, and 10 mm). Specific deformations, material thinning, and forces generated during the process were analyzed. Additionally, the maximum angle of the part’s wall at which the part fails was investigated. For this purpose, AA1050 sheets with thicknesses of 0.8 mm and 1 mm were chosen. These sheets were formed into truncated cone shapes with varying angles, ranging from 40° at the larger base to 75° at the smaller base. The larger base diameter of the truncated cone was set at 85 mm to facilitate deformation measurements using the ARAMIS optical system. The height of the parts at which the 75° angle is reached is 40 mm. The diameter of the tool and the vertical step used were kept constant, with values of 10 mm for the punch diameter and 0.5 mm for the vertical step. The toolpath to form the truncated cone-shaped part was maintained. A summary of the results obtained in this stage is presented in Table 4.

Following the completion of the two experiments and deformation measurements throughout the processing, the following quantities were measured: major strains (ε_1_), minor strains (ε_2_), and material thickness reduction (S_max_). A summary of the measurements obtained through the ARAMIS optical system for the two experiments repeated twice is presented in Table 5. 

The differences in major strains between the two thicknesses of the sheet blanks are not significantly large, except for the minor strains. Thickness reduction shows similar values for both thicknesses of the aluminum alloy. In Figure 7, Figure 8 and Figure 9, it can be observed that the deformations for the AA1050 aluminum alloy are uniformly distributed on the conical wall.

In addition to the deformations measured with the ARAMIS optical system, we also conducted force measurements in the X, Y, and Z directions using the force transducer mentioned in Section 2.3. Table 6 presents the maximum values of real-time measured forces in the three directions. The lowest force values were encountered in the case of trial V3, where a sheet blank of aluminum alloy with a thickness of 0.8 mm was used, as expected. The variation of these forces can be observed in Figure 10.

The parts were successfully manufactured in all four cases and do not exhibit defects, such as material rupture. However, difficulties were encountered in measuring the parts at the end of the processing using the ARAMIS optical measurement system. This is because the angle of the parts’ walls exceeded the visual range of the measurement system. Additionally, it is important to mention that the measured forces do not exceed the permissible load of the robot, which is 2 kN. 

The AA1050 aluminum alloy exhibits a relatively low elongation at fracture, approximately 5%, following the tensile test. However, in the SPIF process, the properties of this alloy are significantly enhanced due to the complex demands that arise within the process. Although the elongation achieved under uniaxial stretching is small, the process allows for an improvement in its properties and the material’s adaptability to process demands. Therefore, AA1050 proves to be a suitable material and can be employed in the studied process without exhibiting the appearance of defects.

### 3.3. Single Point Incremental Forming of Welded Sheets

In the case of the experimental tests presented in this section, where aluminum alloy sheet blanks welded using the WIG process were employed, no piece could be successfully deformed up to the planned height of 25 mm. In comparison, reference sheet blanks made of aluminum alloy were successfully deformed up to a height of 40 mm. Next, deformed parts are presented for cases C1, C5, and C6, along with the area where material failure occurred.

From Figure 11, it can be observed that the failure occurs in the area of the weld bead. This phenomenon is the result of the punch passing over the prominent weld bead, and then upon its return to the flat surface of the sheet blanks, the failure occurs. Each experimental trial was conducted twice to ensure more precise results. The obtained results were analyzed using the Taguchi method, and Table 7 presents the average values for the two measurements taken for each experimental trial, along with the standard deviation, signal-to-noise ratio, and corresponding coefficient of variation.

Out of the nine experimental trials conducted with AA1050 alloy welded sheets using the WIG process, the maximum achieved height was 17 mm in the case of trial C6 (combination of 1–1 mm sheet thicknesses, 0.5 mm vertical step, and 6 mm tool diameter). The minimum height achieved was 15.4 mm in the case of trial C5 (combination of 0.8–1 mm sheet thicknesses, 0.5 mm vertical step, and 10 mm tool diameter).

For the analysis of results, we used the Taguchi method, imposing the “the greater the better” condition to determine the optimal combination of factors leading to the maximum height. Table 8 presents the average signal-to-noise ratio response for the analysis of part height at the moment of material failure.

From the analysis of the influencing factors on the maximum height obtained for the parts, it can be observed that there are optimal levels for the control factors that lead to achieving maximum values. These levels are vertical step at level 1 (S/N = 24.49), tool diameter at level 2 (S/N = 24.36), and combination of sheet blank thicknesses at level 3 (S/N = 24.35). The calculated values for delta and range indicate which of the selected factors have the greatest impact on the studied response. Delta measures the magnitude of the impact by subtracting the smallest average response value of the studied factor from its maximum value. The range provides a ranking of the studied factors, from the one with the highest impact to the one with the least impact on the studied response. The calculated values for delta and range indicate the impact of each factor on the studied response. In the case of part height, the vertical step has the greatest impact (delta = 0.34 and range = 1), followed by the tool diameter (delta = 0.16 and range = 2), while the combination of sheet blank thicknesses shows the smallest impact (delta = 0.11 and range = 3).

Figure 12 and Figure 13 depict the main effects plot and probability plot, allowing us to observe the levels of factors for which the maximum depth of the incrementally deformed welded AA1050 parts through the WIG process will be achieved.

The probability plot represents the probability of obtaining the same results under the same experimental conditions with a confidence level of 95%. In Figure 13 the red squares represents the first set of experiments and the blue circles represent the second set of measurements. The red dash lines and blue lines represents the confidence limits of the 95% probability of obtaining the same results if they are repeated. The main effects plot for the signal-to-noise ratio illustrates the combination of factors for which the highest height is achieved following the incremental formation of the AA1050 welded sheet blanks through the WIG process, namely:Thickness combinations: 1–1 mm;Vertical step: 0.75 mm;Tool diameter: 6 mm.

During the conducted research, it was found that welding sheet blanks end-to-end with butt joints in the I configuration through two passes is a recommended practice. This method is used for sheet blanks made of carbon steels, low-alloyed or stainless steels, aluminum, titanium, and their alloys, due to the significant thermal shock that can occur in the material. However, the possibility of overheating and the occurrence of cracks in the weld bead area should be considered. Additionally, welding sheet blanks with thicknesses less than 1 mm is challenging.

In the experiments carried out according to the experimental design presented in Table 2, all three thickness combinations were welded on both sides to assess the improvement in the behavior of the sheet blanks during single point incremental forming. The experiments were repeated on AA1050 sheet blanks welded through WIG on both sides. Table 9 presents the average values, standard deviation, signal-to-noise ratio, and coefficient of variation for each experimental trial conducted with these sheet blanks.

In the case of AA1050 aluminum alloy sheet blanks welded on both sides using the WIG process, an improvement in behavior was observed, achieving part heights of over 20 mm. The desired depth was successfully achieved in the case of the welded sheet blank thickness combination of 1–1 mm, using a vertical step of 0.75 mm and a tool diameter of 8 mm. Figure 14 presents the major, minor strain and thickness reduction measured using the ARAMIS optical system. In the case of C18, which sustained up to a height of 25 mm, the following results were obtained:For major strain, a value of 80% was obtained;For minor strain, a value of 24.5% was obtained;For thickness reduction, a value of 51.79% was obtained.

Here, the signal-to-noise ratio with the “higher is better” condition was chosen, similar to the case of AA1050 alloy sheet blanks welded on a single side using the WIG process. Table 10 presents the average response of the signal-to-noise ratio for analyzing the height of the pieces at the moment of material failure.

From the Taguchi analysis, optimal levels can be observed to achieve the maximum height of the pieces:The combination of sheet blank thickness at levels 2 and 3 (S/N = 27.02);Vertical step at level 3 (S/N = 27.74);Tool diameter at level 2 (S/N = 27.04).

The greatest impact on the height of the pieces is attributed to the vertical step (delta = 1.52 and rank = 1), followed by the combination of welded sheet blank thickness using the WIG process (delta = 0.29 and rank = 2), and the tool diameter (delta = 0.19 and rank = 3). Figure 12 and Figure 13 depict the graph of main effects for the signal-to-noise ratio and the probability graph.

As a result, the factor with the most significant impact on the height of the pieces is the vertical step, a fact confirmed by Table 10. From the graph of main effects, it can be observed that the maximum heights can be achieved by using combinations of sheet blank thicknesses of 0.8–1 mm and 1–1 mm. Additionally, employing a vertical step of 0.75 mm and a punch with a diameter of 8 mm leads to obtaining the maximum heights for pieces made from AA1050 welded sheet blanks on both sides using the WIG process. From the probability graph, it can be observed that nearly all measurements can be reproduced with a confidence level of 95%.

## 4. Conclusions

Aluminum alloy sheet blanks welded using the WIG process were analyzed comprehensively. Initial research indicated a notable difference in the behavior of these blanks compared to unwelded sheet blanks, particularly in their inability to produce identical parts with depths of 40 mm. This discrepancy led to a critical modification: reducing the piece height to 25 mm to facilitate an in-depth study of the various technological factors influencing them.

A significant portion of the analysis focused on examining the impact of the vertical step and the punch diameter on the AA1050 alloy sheet blanks welded through the WIG process. Despite the efforts, achieving a height of 25 mm for the pieces remained elusive, with the maximum reached height being 17 mm. 

To enhance the deformability of aluminum alloys subjected to WIG welding, a novel proposal suggested the two-sided welding of the aluminum alloy sheet blanks. This approach, tested against cases previously conducted on one-sided welded AA1050 sheet blanks, demonstrated a favorable shift in behavior during the SPIF process, culminating in increased piece depths. Remarkably, the implementation of a 0.5 mm vertical step and a 6 mm punch diameter facilitated the attainment of a 25 mm piece depth with sheet blank thicknesses of 1–1 mm.

In summation, it is plausible to utilize AA1050 aluminum alloy sheet blanks welded on both sides for incremental forming processes, although with a requisite reduction in parts height. 

Potential directions for future research include the exploration of alternative welding techniques subjected to the SPIF process, including the prospective study of friction stir welding as proposed by other researchers. Furthermore, expanding the scope to explore various alloy types, either derived from the same base material or different alloys such as magnesium and titanium, offers a fertile ground for study. Additionally, refining the suitable technological parameters for specific types of welding technology holds great promise for further advancement in this field.

## Figures and Tables

**Figure 1 materials-16-06408-f001:**
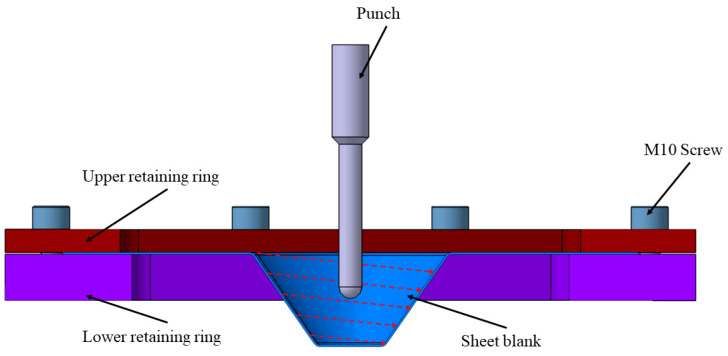
Working principle of SPIF process.

**Figure 2 materials-16-06408-f002:**
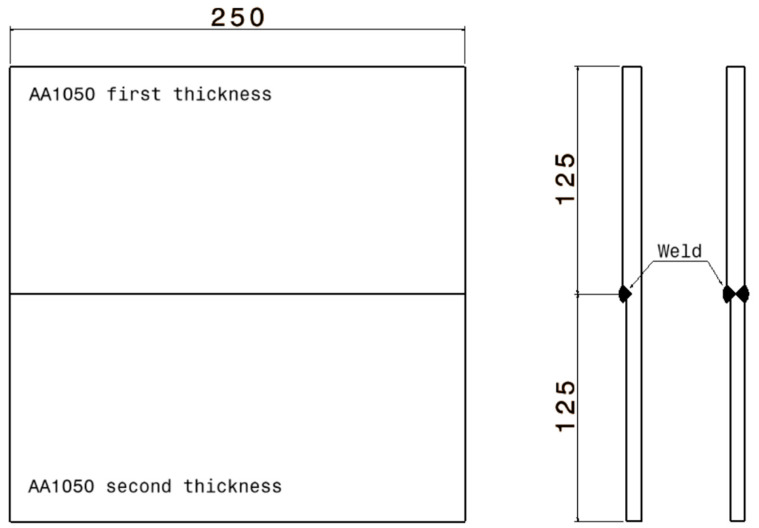
Welding of the specimens with welding bead on one side and on two sides.

**Figure 3 materials-16-06408-f003:**
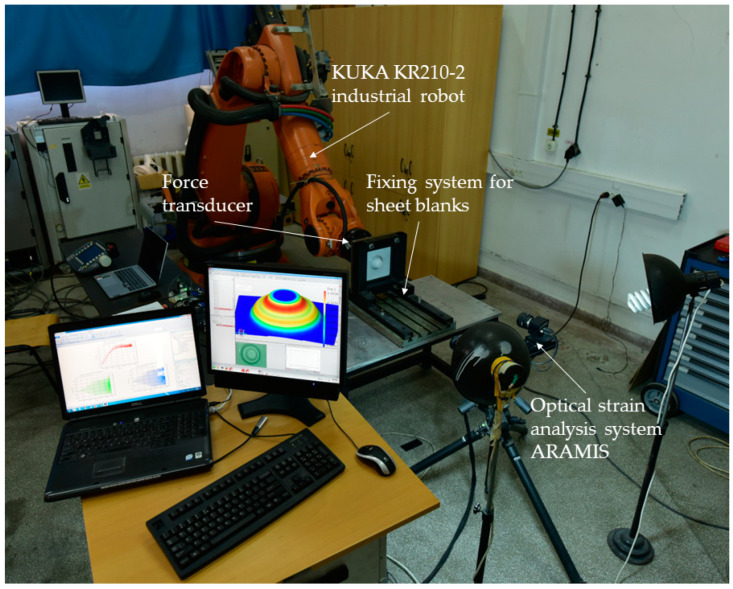
Experimental setup for the study of TWBs processed by SPIF process.

**Figure 4 materials-16-06408-f004:**
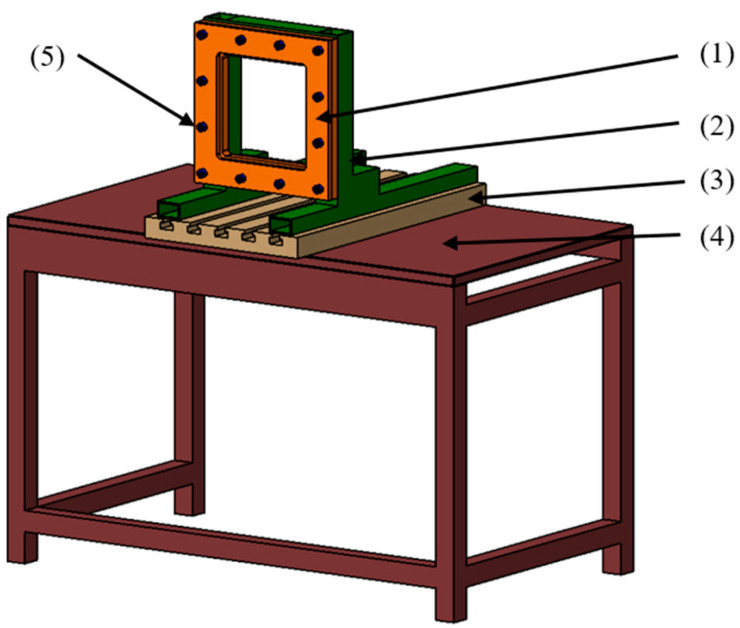
Fixing system for sheet blanks during SPIF process.

**Figure 5 materials-16-06408-f005:**
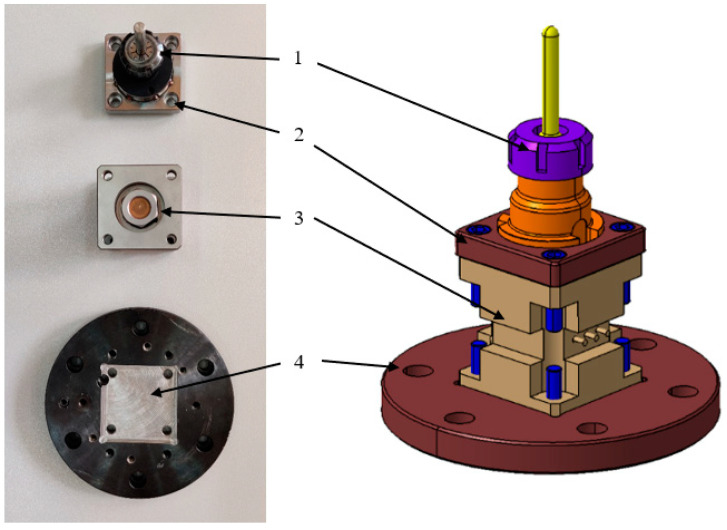
Fixing system toolholder and force transducer.

**Figure 6 materials-16-06408-f006:**
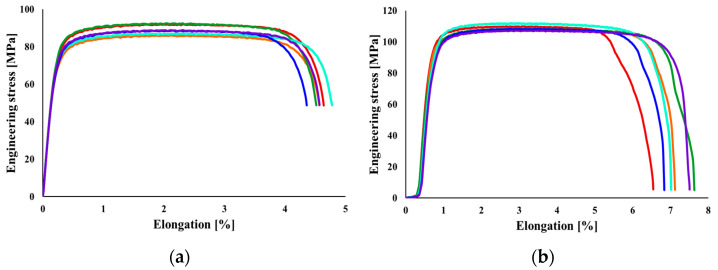
Conventional stress-strain curve for the AA1050 aluminum alloy: (**a**) with a thickness of 0.8 mm and (**b**) with a thickness of 1 mm.

**Figure 7 materials-16-06408-f007:**
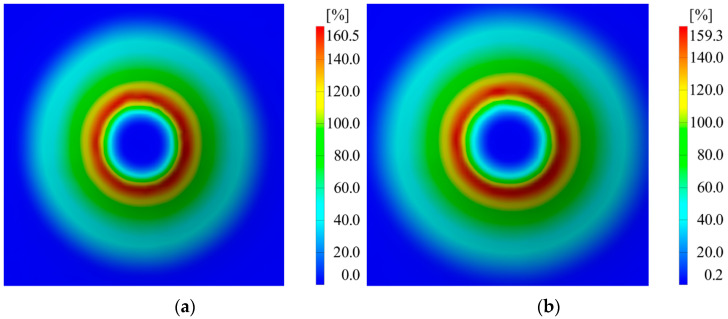
Distribution of major strains in the case of (**a**) V1 and (**b**) V2.

**Figure 8 materials-16-06408-f008:**
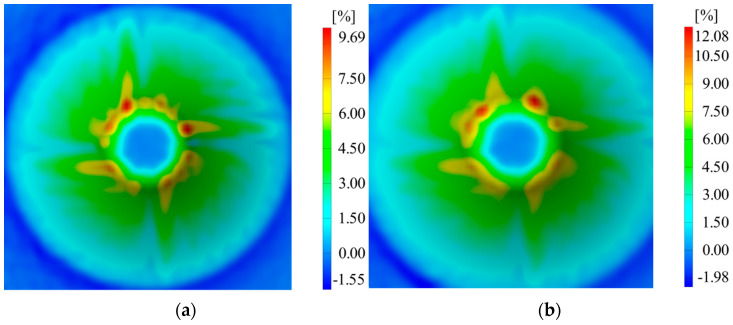
Distribution of minor strains in the case of (**a**) V1 and (**b**) V2.

**Figure 9 materials-16-06408-f009:**
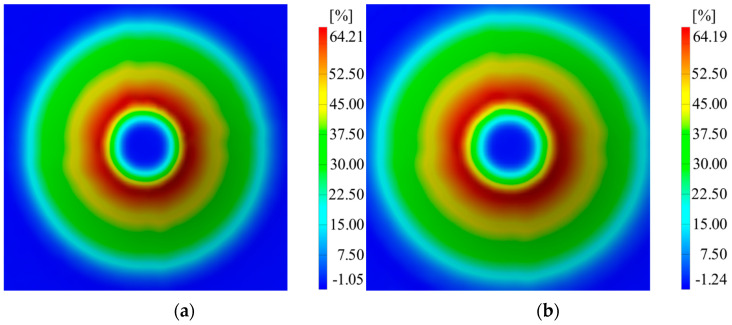
Distribution of thickness reduction in the case of (**a**) V1 and (**b**) V2.

**Figure 10 materials-16-06408-f010:**
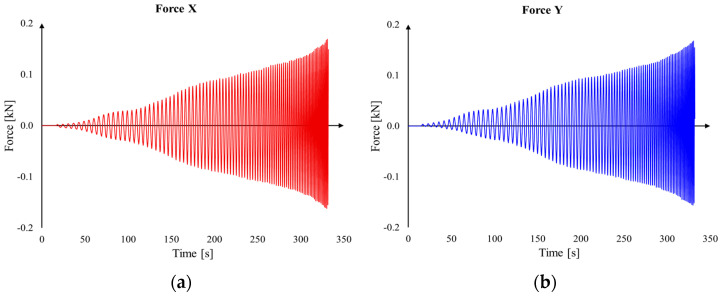
Force variation: (**a**) Fx for case V1, (**b**) Fy for case V1, (**c**) Fz for case V1, (**d**) Fx for case V2, (**e**) Fy for case V2, and (**f**) Fz for case V2.

**Figure 11 materials-16-06408-f011:**
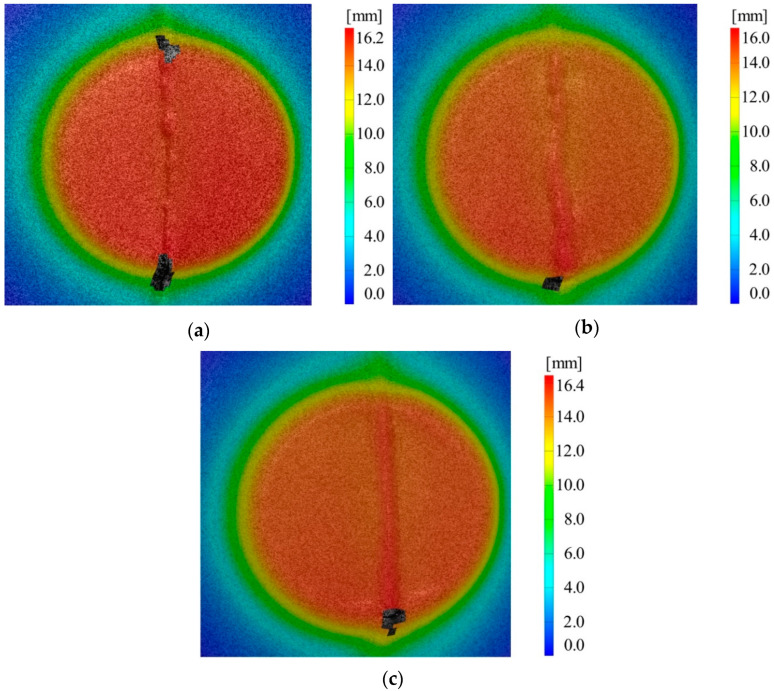
Displacement in the Z direction in the case of (**a**) C1, (**b**) C5, and (**c**) C6.

**Figure 12 materials-16-06408-f012:**
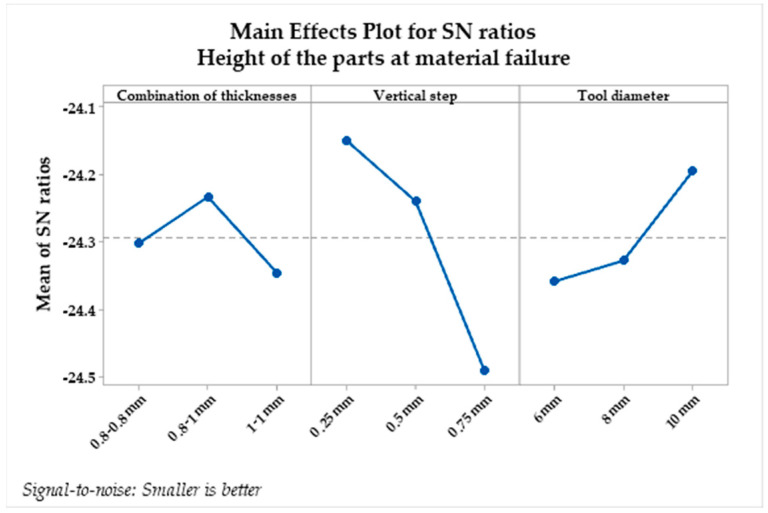
Main effects plot for the signal-to-noise ratio in the case of part height.

**Figure 13 materials-16-06408-f013:**
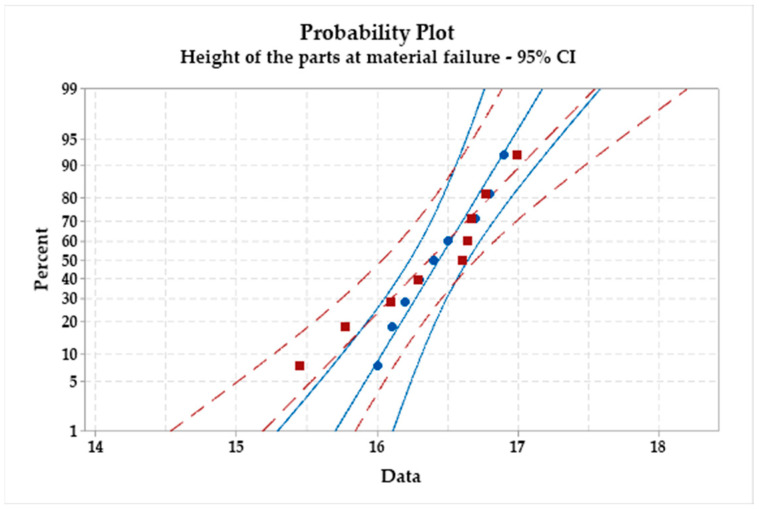
Probability plot for the case of part height.

**Figure 14 materials-16-06408-f014:**
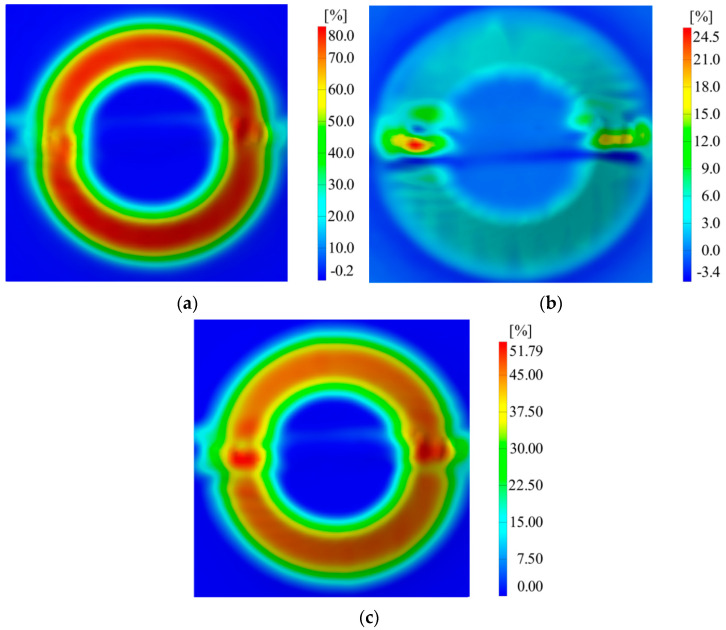
For case C18, the distribution of (**a**) major strains, (**b**) minor strains, and (**c**) thickness reduction.

**Table 1 materials-16-06408-t001:** Chemical composition for AA1050 sheet blanks.

Composition	Al	Mn	Fe	Cu	Mg	Si	Zn	Ti
wt %	99.5	0.05	0.4	0.05	0.05	0.25	0.05	0.03

**Table 2 materials-16-06408-t002:** Experimental planning of AA1050 TWBs welded by WIG using Taguchi method.

Case No.	Combination of Thicknesses [mm]	Wall Angle [°]	Depth of Parts [mm]	Vertical Step [mm]	Punch Diameter [mm]
C1	0.8–0.8	55	25	0.25	6
C2	0.8–1				8
C3	1–1				10
C4	0.8–0.8			0.5	8
C5	0.8–1				10
C6	1–1				6
C7	0.8–0.8			0.75	10
C8	0.8–1				6
C9	1–1				8

**Table 3 materials-16-06408-t003:** Mechanical characteristics of the AA1050 aluminum alloy.

Sheet Thickness [mm]	Specimen Number	E Modulus [GPa]	Yield Strength [MPa]	Ultimate Tensile Strength [MPa]	Strain-Hardening Exponent	Plastic Resistance Coefficient [MPa]	Tensile Strain at Break [mm/mm]
0.8	1.	67.87	85.34	91.97	0.050	114.09	0.046
	2.	64.04	79.53	86.08	0.053	108.55	0.046
	3.	68.61	86.09	92.51	0.051	115.60	0.045
	4.	65.35	80.65	86.91	0.048	106.81	0.048
	5.	64.56	81.95	88.58	0.053	111.30	0.043
	6.	66.82	82.51	88.80	0.049	109.85	0.046
	Mean	66.21	82.679	89.140	0.051	111.035	0.046
	Median	66.09	82.231	88.687	0.050	110.577	0.046
	Standard deviation	1.84	2.579	2.613	0.002	3.335	0.001
	Coefficient of variation	3.41	3.120	2.931	4.324	3.003	2.953
	*p*-value	0.666	0.602	0.436	0.885	0.868	0.667
1	1.	70.87	101.44	109.88	0.067	144.44	0.065
	2.	69.85	99.88	107.68	0.057	135.60	0.071
	3.	68.75	98.85	108.48	0.061	138.30	0.076
	4.	69.75	103.62	111.83	0.071	149.20	0.070
	5.	66.69	99.54	108.19	0.076	147.19	0.068
	6.	70.88	101.44	109.88	0.067	144.44	0.065
	Mean	69.46	100.337	108.889	0.065	141.756	0.071
	Median	69.80	99.709	108.332	0.064	141.373	0.071
	Standard deviation	1.57	1.885	1.696	0.008	5.957	0.004
	Coefficient of variation	2.48	1.878	1.558	11.517	4.202	5.683
	*p*-value	0.265	0.223	0.288	0.551	0.303	0.785

**Table 4 materials-16-06408-t004:** Initial experiments conducted on non-welded incrementally deformed sheets.

Experimental Test	Sheet Thickness [mm]	Wall Angle [°]	Part Height [mm]	Vertical Step [mm]	Tool Diameter [mm]
V1	0.8	Variable, ranging from 40° to 75°	40	0.5	10
V2	1				

**Table 5 materials-16-06408-t005:** The values of major, minor strains and thickness reductions for the initial experimental test.

Experimental Test	Major Strains		Minor Strains	
	ε_11_ [%]	ε_12_ [%]	ε_21_ [%]	ε_21_ [%]
V1	160.5	159.5	9.69	9.65
V2	159.3	164.2	12.08	12.62
-	**Thickness reduction**			
	$S_{max1}$ **[%]**		$S_{max2}$ **[%]**	
V1	64.21		63.75	
V2	64.19		65.49	

**Table 6 materials-16-06408-t006:** The maximum values of the three components of the forces.

Experimental Test	$F_{x}$		$F_{y}$		$F_{z}$	
	$F_{x1}$ [kN]	$F_{x2}$ [kN]	$F_{y1}$ [kN]	$F_{y2}$ [kN]	$F_{z1}$ [kN]	$F_{z2}$ [kN]
V1	0.190	0.190	0.168	0.167	0.291	0.289
V2	0.226	0.240	0.200	0.209	0.405	0.426

**Table 7 materials-16-06408-t007:** Analysis of part height at the moment of material failure.

Experimental Test	Height of the Part at the Moment of Material Failure [mm]			Standard Deviation	Signal-to-Noise Ratio	Coefficient of Variation
	$H_{1}$	$H_{2}$	$H_{mean}$			
C1	16.2	15.8	15.983	0.306	24.071	0.019
C2	16.4	16.7	16.533	0.188	24.366	0.011
C3	16.8	16.6	16.719	0.114	24.464	0.007
C4	16.1	16.3	16.193	0.131	24.186	0.008
C5	16.0	15.4	15.724	0.390	23.927	0.025
C6	16.9	17.0	16.946	0.064	24.581	0.004
C7	16.3	16.1	16.197	0.146	24.188	0.009
C8	16.5	16.8	16.633	0.188	24.418	0.011
C9	16.7	16.6	16.651	0.069	24.429	0.004

**Table 8 materials-16-06408-t008:** The average signal-to-noise ratio response for the part height at the moment of material failure in case of one-sided welded blanks.

Level	Combination of Thicknesses	Vertical Step	Tool Diameter
1	24.30	24.15	24.36
2	24.23	24.24	24.33
3	24.35	24.49	24.19
Delta	0.11	0.34	0.16
Range	3	1	2

**Table 9 materials-16-06408-t009:** Analysis of the height of double-sided welded parts at the moment of material failure.

Experimental Test	Height of the Part at the Moment of Material Failure [mm]			Standard Deviation	Signal-to-Noise Ratio	Coefficient of Variation
	$H_{1}$	$H_{2}$	$H_{mean}$			
C10	20.1	19.5	19.7973	0.235	26.388	0.011
C11	21.6	22.4	21.9942	0.213	26.788	0.010
C12	23.6	23.5	23.5332	0.304	27.893	0.012
C13	20.7	21.0	20.8662	0.505	26.329	0.024
C14	22.0	21.7	21.8492	0.140	26.848	0.006
C15	24.6	25.0	24.8146	0.291	27.886	0.012
C16	21.1	20.4	20.7326	0.235	26.388	0.011
C17	21.9	22.1	21.9990	0.213	26.788	0.010
C18	25.0	24.6	24.7939	0.304	27.893	0.012

**Table 10 materials-16-06408-t010:** The average signal-to-noise ratio response for the part height at the moment of material failure in case of two-sided welded blanks.

Level	Combination of Thicknesses	Vertical Step	Tool Diameter
1	26.73	26.22	26.89
2	27.02	26.83	27.04
3	27.02	27.74	26.85
Delta	0.29	1.52	0.19
Range	2	1	3

## Data Availability

The data presented in this study are openly available.

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
