# Peer review of "Experimental Research on Wolfram Inert Gas AA1050 Aluminum Alloy Tailor Welded Blanks Processed by Single Point Incremental Forming Process"

_materials, 2023, doi:10.3390/ma16196408_

Round 1

Reviewer 1 Report

This manuscript investigates the behaviors and properties of AA1050 aluminum alloy through SPIF. However, after careful evaluation, I regret to inform you that I cannot recommend this manuscript for publication due to several concerns. Primarily, while the study delves into AA1050 alloy behavior, its scientific depth and novelty do not align with the focus of this journal. Regrettably, I did not derive significant insights from the work, which tends to resemble a data-centric report. The manuscript would benefit greatly from a more robust emphasis on introducing innovative findings and contributions to the field. Furthermore, enhancing scientific clarity and grammar is essential to improve readability and relevance. The manuscript title, "Wolfram Inert Gas AA1050 Alloy," is ambiguous and requires clarification. Similarly, the introduction requires revision as the lengthy description of other researchers' work without your analysis diminishes its significance. Moreover, the motivation driving your research lacks effective highlighting. Lastly, figures 1 and 2 appear to incorporate images from other researchers' work. To maintain ethical standards, obtaining proper authorization for the reproduction of these images is crucial.

The language used in the manuscript presents challenges in clarity and comprehension, making it hard to follow. Consider simplifying sentence structures and enhancing clarity by avoiding unnecessary jargon. Additionally, proofreading for grammar and punctuation will help improve overall readability. Organizing content logically and seeking input from peers can also enhance the manuscript's effectiveness.

Author Response

Dear Reviewer,

Thank you for your meticulous review and valuable feedback on our manuscript. We truly appreciate the time and effort you have invested in evaluating our work. In response to your observations, we have implemented several substantial changes to enhance the scientific depth and clarity of the manuscript. 

Reviewer 2 Report

Dear Author(s), the manuscript ‘Experimental Research on Wolfram Inert Gas AA1050 Aluminum Alloy Tailor Welded Blanks processed by Single Point Incremental Forming process’, Manuscript ID: materials-2609065, have some weakness that must be revised suitably.

Please find below some, of the most significant issues:

1.      Some introducing words to the area (field) of study should be added to the ‘Abstract’ section. Then, as presented, Author(s) should provide some information on the analysis assigned.

2.      Even one sentence must be provided on the advantages and disadvantages of the studies performed. Currently, Author(s) add some profilts only. Indicating some of the limitation can be useful for ‘The Outlook’.

3.      There is no critical review in the ‘Introduction’ section, respectively, the main dsadvantages of the previous tudies, presented in lines 115-122 is not efficient. The lack of the current state of knowledge must be emphasized that makes the motivation more clear. From that matter, the motivation seems to be hidden from the full understanding for the Reader(s).

4.      Generally to the ‘Introduction’ section, it is interesting and well written, however, is a little bit long. Please try to quantify the main purpose and the literature review that makes the motivation clear.

5.      In the section 2, subsection 2.1 or 2.2, the flow chart of the procedure must be presented. Currently, for a regular reader, it is difficult to retrieve the main methodology. Figure 2 from ‘Introduction’ is not sufficient.

6.      In section 2.1, the sentences from lines 191-207, are more like for the ‘introduction’ than presenting the methodology. I suggest to remove those sentences and mention only those influencing the main studies. Author(s) present too many irrelevant information.

7.      Some value described in section 2.2 are not justified and looks like seltected arbitrarily. Author(s) should justify their values or reference to the proper sources where differencies were comprehensively studied.

8.      For the section 3, considering all of the subsections (3.1, 3.2 and 3.3) there is no proper critical discussion. From the current form it is difficult what the Author(s) are trying to convey as a major study.

9.      The advantages and disadvantages (especially limitations) of the studies should be discussed in the section 3. Similar to the ‘Abstract’ section, there is no words on the limitations that can define any further prospects of the study.

10.  The ‘Conclusion’ section should be divided for a separate and numbered gaps. In the current form, the main novelty is hidden from the other proposals. Morevoer, the detailed comments (with parametric description) should be separated from those general.

From the above, the reviewed manuscript must be improved appropiately before any further processing, if allowed by the Editor.

Reviewer 3 Report

Comments:

ABSTRACT:

Starting from Line 17, the following sentence is unclear:  The main conclusion of the paper is that during SPIF process, fracture of the parts occurs at the one-sided welded blanks, but in the case of two-sided welded blanks it is possible to reach certain depth of parts by using the right technological parameters for welding process.”

Some suggestions

1)       This is a too long sentence, break it in more simpler blocks.

2)       Report in the last sentences of the abstract a brief summary of the main results and then, if the case, but it is not mandatory, the conclusion.

3)       Avoid general expressions such us: “…it is possible to reach certain depth of parts…” What means a “certain depth”…please be precise and explain clearer your conclusion. Probably the abstract is not the right place.

Line 27: It is suggested to add a scheme for a simpler understanding of the SPIF process.

Line 49-63 and Fig.1: these are literature results or new results? It seems that You have reported what is already achieved in [7]. Fig.1 is already reported in literature, so it is not required here, but if the Authors needs, it is mandatory obtaining a permission to report the Fig.1 and results that are covered by Copyright. It is not enough quote the reference.

 General comment: in the Introduction all the discussion (line 25-168) is a summary of literature data and the presentation of actual research starts at line 169 -173…only  five lines for presenting  the experimental work? This is acceptable for a student thesis or a technical report, but this is a scientific Journal. The introduction should clearly describe the scope, (it is well ok) the experimental tests (in advance and this point is a weak point) and all the methodology and the aim of the paper.

 Move line 175 -182 in the introduction and also in a more synthetic way in the abstract.

 Tab.3: All the E (Young Modulus) are out of the literature range. According to my experience the E modulus, should be expressed in GPa not in MPa, but the fact is that the value is ranging around 68-70 GPa not less than the half.

It is your measurement chain (extensometers) at half or full “Wheatstone bridge”? Check please the right selection.

The rest of the paper seems well written. I stop here the review it is enough for a "major"

The paper presents a mixture of "technical-report style" and good experimental data, but it is confused with long sentences.

Round 2

Reviewer 1 Report

I've noticed a marked improvement in this article. However, there are essential modifications required, particularly in the introduction. Please consider the following questions and suggestions.

1. From lines 30 to 42, you've provided a commendable and succinct summary highlighting the current challenges and limitations of SPIF. To strengthen your arguments, please incorporate a few references. For instance, in line 32 where you mention, "the SPIF tends to be slower compared to other methods of plastic deformation," could you specify a few of these other methods and cite relevant references?

2. In lines 33 and 34, there's no need for a hyphen within words like 'predom-inantly' and 'cre-ating.' Please correct this.

3. Between lines 61 to 75, the study by Ambrogio can be more succinctly described. Consider integrating this into the previous paragraph as supporting evidence for the initial stages of single point incremental forming of welded sheets, emphasizing its role in the development of single point incremental forming.

4. From lines 76 to 95, I found the content a bit difficult to follow. If your intention is to continue discussing the development of SPIF, perhaps start with a brief summary outlining the merits and challenges at this current stage. Subsequently, you can incorporate the referenced work. The section seems quite detailed; if the focus is on certain techniques or designs and their respective results, I suggest highlighting them more prominently, possibly in your introduction..

5. From lines 117 to 131, you've provided a commendable example with Alinaghian's work. However, it feels a bit redundant, especially since it's just one example among others in the introduction. It might be beneficial to streamline this section, summarizing other people's work more concisely.

6. Having reviewed the introduction, I must stress the need for significant improvements. I get the sense that the authors intend to concentrate on the evolution of SPIF, particularly the techniques. I'd suggest structuring the introduction to highlight the progression of more advanced techniques or significant achievements in parameters. Furthermore, please remove any details that don't align with the overarching theme of development. Overloading the section with numerous studies can overwhelm and confuse the reader.

7. Within the motivation section, it would be impactful if the author could provide a tangible application that requires sheet blanks of the same material but with varying thickness (as referenced in line 221). This would underscore the novelty of the manuscript more convincingly, rather than just indicating that others haven't explored this area.

8. What is the unit used in Table 1? Please specify the unit both within the table itself and in the surrounding text. Ensure that all tables in the manuscript clearly state their units. Additionally, please also make necessary revisions to Table 8 and Table 10.,

9. Please label the components within the figure, like the 'fixing system,' 'robot,' 'force transducer,' and so on, similar to the annotations in Figure 4.

10. Lines 373 to 381 seem more appropriate for the experimental section rather than the results and discussion. Please relocate this content to its relevant section..

11. For lines 425 to 429, there's no need to outline what will be covered in this section. You can directly present your findings. Keep in mind, you're in the "results and discussions" section; detailed procedural explanations belong in the experimental section.

Reviewer 2 Report

The Author(s) improved the submission appropiately and responded all of the raised issues suitably that the paper can be now considered for the acceptance.

Author Response

Dear reviewer, 

We sincerely appreciate your time and effort in reviewing our manuscript and for recognizing the improvements made. We are elated by your positive remarks regarding the revisions and your recommendation for the acceptance of our paper.

Thank you.

Reviewer 3 Report

Thank You for having dramatically improved the paper.

Author Response

(The authors gave the same response as above.)
